# Injectable Human Hair Keratin–Fibrinogen Hydrogels for Engineering 3D Microenvironments to Accelerate Oral Tissue Regeneration

**DOI:** 10.3390/ijms222413269

**Published:** 2021-12-09

**Authors:** Hyeon Jeong Kang, Nare Ko, Seung Jun Oh, Seong Yeong An, Yu-Shik Hwang, So Yeon Kim

**Affiliations:** 1Department of Maxillofacial Biomedical Engineering and Institute of Oral Biology, School of Dentistry, Kyung Hee University, Seoul 02447, Korea; khj001@khu.ac.kr (H.J.K.); syan0426@gmail.com (S.Y.A.); 2Biomedical Research Center, Asan Institute for Life Sciences, 88 Olympic-ro 43-gil, Songpa-gu, Seoul 05505, Korea; nare.ko@amc.seoul.kr; 3Department of Nuclear Medicine, Asan Medical Center, University of Ulsan College of Medicine, 88 Olympic-ro 43-gil, Songpa-gu, Seoul 05505, Korea; sjoh@amc.seoul.kr; 4Department of Dental Hygiene, College of Health & Medical Sciences, Cheongju University, Cheongju 28503, Korea

**Keywords:** human hair keratin, fibrinogen, injectable hydrogel, human gingival fibroblast, biomaterials, biocompatibility

## Abstract

Traumatic injury of the oral cavity is atypical and often accompanied by uncontrolled bleeding and inflammation. Injectable hydrogels have been considered to be promising candidates for the treatment of oral injuries because of their simple formulation, minimally invasive application technique, and site-specific delivery. Fibrinogen-based hydrogels have been widely explored as effective materials for wound healing in tissue engineering due to their uniqueness. Recently, an injectable foam has taken the spotlight. However, the fibrin component of this biomaterial is relatively stiff. To address these challenges, we created keratin-conjugated fibrinogen (KRT-FIB). This study aimed to develop a novel keratin biomaterial and assess cell–biomaterial interactions. Consequently, a novel injectable KRT-FIB hydrogel was optimized through rheological measurements, and its injection performance, swelling behavior, and surface morphology were investigated. We observed an excellent cell viability, proliferation, and migration/cell–cell interaction, indicating that the novel KRT-FIB-injectable hydrogel is a promising platform for oral tissue regeneration with a high clinical applicability.

## 1. Introduction

Keratin (KRT) is an insoluble protein that forms a part of intermediate filaments in epidermal appendageal structures, hair, nails, horn, hoofs, wool, and feathers [1]. KRT serves important structural and protective functions, particularly in the epithelium [2,3]. In particular, human hair-derived KRTs contain excellent cell adhesion motif Leu-Asp-Val (LDV) that enables the extracellular substrate for cell attachment and support [4,5]. We have recently reported that human hair-derived KRT is highly attractive for wound healing therapy because of its inherent bioactivity, biocompatibility, and physical properties [6]. Despite these advantages, previous KRT-based hydrogel studies are only limited to wool- or feather-based KRT proteins [7]. In addition, hydrogels based on human hair-derived KRTs, especially injectable platforms, have never been studied for oral tissue regeneration.

Fibrinogen (FIB) is a promising candidate due to its innate wound healing process that stimulates reparative cell activity, as well as exhibits both angiogenic and anti-inflammatory properties that are essential for wound healing and tissue repair [8,9,10]. In addition, FIB is known as a key protein for the regulation of angiogenesis and bone regeneration. The first clinically approved FIB hydrogel for tissue repair is produced by the formation of FIB polymers between transglutaminase factor XIII and calcium ions in the presence of thrombin (THR) [11]. However, current FIB hydrogels exhibit weak mechanical strength with a lack of porosity and adhesive properties due to their limited design ability of the protein backbone [12]. Hydrogels comprising FIB alone are known to increase scaffold contraction and do not support continuous cell growth [13]. Thus, such FIB hydrogels are relatively stiff and difficult to handle when implanted.

To resolve these problems, we developed a novel injectable KRT-FIB hydrogel (KFH) for oral tissue regeneration (Figure 1). To achieve the homogeneous formation of the KFH, human hair-driven KRT proteins were covalently conjugated with the FIB proteins via a facile coupling reaction. Importantly, the resulting KRT-FIB precursor is the first example of KRT-conjugated FIB material that enables the formation of well-established KFH in the presence of THR. With the optimal ratio of KRT: FIB, KFH demonstrated great injectable performance, swelling behavior, and high porosity. Human gingival fibroblasts (HGFs) encapsulated in the KFH showed excellent cell viability, proliferation, and migration with cell-cell interaction. These results revealed that the KFH has a great potential to promote oral tissue regeneration by the formation of the HGFs network involved in the basal layer of the oral gingival dermis. Moreover, the injectable KFH provides high clinical applicability with reducing reduce patient pain, treatment cost, and defect recovery time.

## 2. Results and Discussion

### 2.1. Synthesis of KRT-FIB Precursors

Although both KRT and FIB are hydrophilic proteins, they cannot be integrated into homogeneous hydrogels in the presence of THR. To address this challenge, covalently linked KRT-FIB precursors were synthesized via a facile coupling reaction (Figure 2a). The human hair-driven KRT protein was first succinylated to introduce carboxylic acids as terminal functional groups. The carboxyl groups on the resulting succinyl KRT protein (KRT-COOH) enable conjugation to the amine groups on FIB via a carbodiimide/N-hydroxysuccinimide (EDC/NHS) reaction. With varying molar ratios of KRT and FIB, a series of precursors were prepared, abbreviated as KF-1, KF-3, and KF-6, which stand for the 1:1, 3:1, and 6:1 mol ratios of KRT: FIB. Such direct conjugation of KRT and FIB facilitates the formation of well-defined KFH without phase separation. Sodium dodecyl sulfate-polyacrylamide gel electrophoresis (SDS-PAGE) was performed to study the change in molecular weight of the KRT-FIB precursors during the reaction (Figure 2b). The KRT protein weighs less than 40 kDa and is not shown on the gel, and the band of the FIB protein appears at 350 kDa. After conjugation, the molecular weight of KRT significantly increased and thick bands corresponding to the KRT-FIB precursors were positioned higher than 350 kDa. Note that precursors larger than 500 kDa were not capable of passing through the polyacrylamide gel and were positioned at the top of the stacking gel, indicating the formation of precursors with a molecular weight larger than 500 kDa.

Furthermore, the effect of KRT conjugation was evaluated by analyzing its thermal properties using thermogravimetric analysis (TGA). Figure 2c shows the TGA data of the weight loss of the KRT-FIB precursor upon heating, compared with the KRT and FIB proteins as controls. For the precursor, major weight loss started at 259 °C. This temperature is lower than that for the KRT protein (266 °C), but higher than that for the FIB protein (202 °C, the first curve), indicating that the introduction of KRT enhances the thermal stability of FIB for the formation of hydrogels.

### 2.2. Preparation and Rheological Studies of KFHs

The gelation rate is a critical parameter for hydrogels for injectable applications. Slow gelation causes the lateral spreading of precursors to non-target sites, whereas fast gelation clogs needles due to the formation of entanglements in the syringe [14,15] To demonstrate rapid gelation with a high efficiency, KFHs were prepared via THR-induced crosslinking (Figure 3a). In the presence of THR, FIB proteins are degraded into fibrin monomers by the cleavage of fibrinopeptide A/B. N-terminal fragments of α chains (known as “knobs”) on the fibrin monomers bind to the complementary C-terminal of the β/γ chains (known as “holes”) and the fibrin monomers are polymerized via intermolecular interactions. Such THR-mediated knob-hole interactions formulate the 3D network structure of the KFH, crosslinked by fibrin polymers (Figure 3b). A series of KFHs with varying KRT: FIB mole ratios were prepared (Figure 3c). In the presence of 5 unit/mL of THR, the gelation of KRT-FIB precursors occurred spontaneously and was completed within 1 min.

To achieve clinically acceptable injectability, hydrogels must be designed considering several critical physical properties: (1) shear-thinning behavior [16], (2) sufficient strength to resist deformations [17] and (3) a higher frequency-dependent storage modulus (G’) than the corresponding loss modulus (G”) [18]. To ascertain mechanical properties for injectable applications, rheological measurements of KFHs (1, 3, and 6) were carried out. First, the change in the complex viscosity of the KFH was measured with an oscillatory frequency sweep at 37 °C. Complex viscosity is an important parameter to consider in the design of injectable platforms, as it measures the ability of the hydrogels not only to respond to changes in shear stress during the injection but also to resist deformation within a tissue after injection. As depicted in Figure 4a, the hydrogels show a continuous decrease in complex viscosity with an increasing frequency sweep, which indicates shear-thinning viscoelastic behavior. In addition, the complex viscosity of the KFH gradually increased with the increasing molar ratio of KRT to FIB. This result reveals that the KFHs are suitable for injection through needle extrusion and that the viscosity can be regulated by the KRT content [18].

To further study the elastic and viscous behaviors, the G’ and G” moduli of the KFHs were examined (Figure 4b–d). Constant G’ and G” values and low tan δ (the ratio of G”/G’; tan δ <1) revealed the gel-like behavior of the KFH. As the G’ value was proportional to the KRT content with increasing stiffness, the KFH-3 hydrogel fabricated with the KF-3 precursor (mole ratio of KRT: FIB = 3:1) was selected and used for further studies. These rheological results suggest that KFHs have great potential as injectable platforms.

### 2.3. Injectable Performance, Swelling, and Degradation Behavior of KFH-3

Excellent injectable performance and swelling behavior of a hydrogel are important characteristics to facilitate the filling of irregularly shaped defects, simple formulation, non-invasive technique, and site-specific action [19,20]. Figure 5a indicates that the KFH-3 hydrogel passed smoothly through an 18 G needle and immediately acquired a gel-like structure. Such an excellent injectable performance indicates that the KFHs facilitate the filling of irregular defects. In addition, the swelling ratio of the KFH-3 hydrogel was almost six times higher than that of the FIB-H hydrogel (Figure 5b). As well-defined hydrogels do not dissolve but swell in water, this result revealed that the swelling rate of the hydrogels was significantly improved by the introduction of KRT. Such water absorption ability and swelling kinetics of hydrogels can be modulated by porosity [21]. Scanning electron microscopy (SEM) images show that KFH-3 has a highly porous structure with a 10–100 μm diameter compared to the non-porous FIB-H (Figure 5c). These results indicate that the swelling ratio of the KFH-3 strongly depends on the porous morphology of the hydrogel.

The in vitro degradation behavior of KFH-3 and FIB-H hydrogels in warm cell media (50 °C) at pH 5.5 was monitored by a comparison of the weight change. As shown in Figure 5d, the weight of both hydrogel samples steadily decreased. In addition, a significant difference in weight between KFH-3 and FIB-H hydrogels was observed. After 12 h of incubation, the weight of KFH-3 hydrogel decreased to 20% and that of FIB-H decreased to 53.3%. This result indicated that KRT-conjugated fibrin-based hydrogel was more resistant to high-temperature and acidic conditions than pristine fibrin-based hydrogel, protecting itself from the integrity loss.

### 2.4. HGF Viability in KRT-Based Fibrin Hydrogels: Cytotoxicity Studies and 3D Cell Encapsulation in KRT-Based Fibrin Hydrogels

Porosity is a property of hydrogels that is important for tissue regeneration [22,23]. Porous hydrogels provide the necessary space for cell growth and vascularization [24,25,26]. It is well-known that such a three-dimensional (3D) structure of hydrogels is necessary to achieve successful oral tissue regeneration because it can be used as a delivery vehicle for bioactive substances in cells [22]. The efficient transportation of nutrients and oxygen through interconnected pathways promotes cell proliferation, migration, and cell–cell contact [27,28].

The cytotoxicity analysis of cell adhesion and proliferation was performed using HGFs for the LIVE/DEAD (Figure 6a,b) and CCK-8 assay (Figure 6f). The cell proliferation of all groups increased with the culture time. Furthermore, except for the samples from day 1, much higher cell viabilities in the KFH-3 were detected than those of the FIB-H at the other time intervals, indicating better cytocompatibility to support cell proliferation. In addition, we observed that HGFs proliferated and exhibited spreading morphology in encapsulated KFH. Phalloidin staining (Figure 6c,d), used to perform cellular interaction studies using HGF cells, demonstrated the significant free flow of biological fluids and cell migration and growth inside the KFH.

KFH-3, possessing its inherent interconnected porous structure, showed a high cell proliferation and cell viability, while the non-porous FIB-H showed a lower cytocompatibility than KFH-3. HGFs elongated and spread out to form interconnected networks in the KFH-3 hydrogel, compared with FIB-H. Notably, hydrogels with small pores of less than 10 μm are undesirable for use in tissue engineering applications, as they limit cell migration and nutrient diffusion [29,30,31,32]. FIB-H has small pores and large, non-uniform, and different surface areas. After 3 days of culture, the live cell density in the hydrogel with KRT was also higher than that in the hydrogel without KRT. Thus, KFH-3 induced a higher cell proliferation and exhibited a larger adhesion area than the FIB-H hydrogel. The hydrogel developed in this study has the porosity and microarchitecture required for use in an in vitro 3D environment, allowing the free flow of biological fluids and cell migration and growth inside the material.

## 3. Materials and Methods

### 3.1. Materials

The KRT protein was extracted from human hair and the detailed procedure has been described in our previous study [33]. Human plasma FIB (Merck Millipore, MA, USA) and human plasma THR (Calbiochem, CA, USA) were used to prepare the fibrin gel. Dulbecco’s phosphate-buffered saline (PBS) was obtained from Thermo Fisher Scientific (Waltham, MA, USA). EDC/NHS, Tris-buffered saline, and sodium hydroxide (NaOH) were obtained from Sigma-Aldrich.

HGFs were purchased from ScienCell Research Laboratories (Carlsbad, CA, USA). All chemicals were used without further purification.

### 3.2. Succinylation of KRT Protein (KRT-COOH)

Succinic anhydride (25 mg) dissolved in PBS was added dropwise to a solution of KRT (200 mg) in 100 mL PBS (pH 6–7). The resulting mixture was stirred at room temperature for 1 h and distilled water (DW) for 4 h. The purified KRT-COOH was lyophilized and stored at −20 °C until further use.

### 3.3. Synthesis of KRT-FIB Precursors via An EDC/NHS Coupling Reaction

To synthesize a series of KRT-FIB precursors, different amounts of KRT-COOH (0, 5, 10, 30, and 60 mg) were activated in the presence of EDC/NHS. First, 19.25 mg of EDC was added to a solution of KRT-COOH in 100 mL of PBS. Then, 54.25 mg of sulfur-NHS was added to the solution and stirred at room temperature for 15 min. Then, 62.5 μL of 2-mercaptoethanol was added drop-wise. Subsequently, the pH of the resulting mixture was adjusted to 7.0 via the addition of 10X PBS (10 mL), and 100 mg of FIB was added to the mixture while stirring at room temperature. The reaction was allowed to proceed for 2 h and was then quenched by the addition of a Tris buffer. The pH of the quenched solution was increased to 8.0 by adding 1 N NaOH. The resulting precursor solutions were placed in a dialysis membrane with a molecular weight cut-off of 25,000 g/mol (Spectrum Laboratories Inc., Rancho Dominguez, CA, USA) and dialyzed over DW for three days. The purified precursors were then lyophilized and stored at −20 °C until further use.

### 3.4. SDS-PAGE

SDS-PAGE was performed to confirm the formation of the precursors. All samples (KF-1, 3, and 6) were run on a 7% Tris-acrylamide running gel with 4% stacking, for 90 min at 150 V, and the electrophoretic bands were stained with Coomassie Brilliant Blue R-250.

### 3.5. Preparation of KFHs and FIB-H Hydrogels

A series of KFHs (1, 3, and 6) were prepared using the corresponding precursors with different KRT contents (KF-1, 3, 6). First, solutions including each precursor (20 mg/mL), THR (5 IU/mL), and CaCl_2_ (2.5 mg/mL) were prepared. Then, 2 mL of the precursor solution was loaded onto one side of a dual-chambered syringe with an 18 G needle (diameter = 1.3 mm), and 0.1 mL of THR solution was loaded onto the other side. The pre-filled syringe was slowly extruded on a glass plate at room temperature. As a control, FIB-H was prepared following the same procedure as the FIB solution (20 mg/mL). To evaluate the injection performance, the solutions of the precursor and THR prefilled in the dual-chambered syringe were injected into DW and on a glass plate.

### 3.6. Characterization of Injectable Hydrogels

The thermal properties of a series of KRT-FIB precursors and the corresponding hydrogels (KFHs) were determined by TGA (SDT Q600, TA Instruments, New Castle, PA, USA) under nitrogen gas flow, and the temperature ranged from 20 to 600 °C at a heating rate of 10 °C/min. The morphology of the hydrogels was observed by SEM (S-4700, Hitachi, Japan) with a gold coating. The rheological properties of the hydrogels were investigated using an ARES-G2 rheometer (TA Instruments, New Castle, PA, USA) at 37 °C with a parallel-plate geometry of 8 mm in diameter

To examine the injectable performance of the KFH, an extrusion experiment was performed. A dual-chambered syringe was used to make injectable hydrogels. In all cases, the precursor solutions/THR in each dual-chambered syringe were liquids with very low viscosity and were easily extruded, first through a static mixer placed at the outlet of the dual-chambered syringe and then through an 18 G needle. A dual-chambered syringe loaded with a FIB and KF-3 precursor, and THR solution was slowly extruded by hand into DW and on a glass plate. The extrusion was carried out by pressing down on the syringe plunger, by hand. The rheology of the injectable hydrogels was tested using a rheometer (HAAKE, Model MARS, Karlsruhe, Germany). Rheology Advantage Data Analysis (TA Instruments, Karlsruhe, Germany) software was then used to plot the frequency sweeps. All hydrogels were cylindrical with a diameter of 20 mm and a thickness of 1 mm. A dynamic frequency sweep test from 0.1 to 100 rad/s was performed to determine the dynamic storage modulus (G′) of each hydrogel, at a strain rate confirmed to be in the linear viscoelastic range for each type of hydrogel prepared. The temperature was maintained at 37 °C during all measurements. At least five different hydrogels were tested with the same experimental settings; average values are presented.

### 3.7. Swelling and Degradation Studies

The swelling ratios of KFH-3 and FIB-H were evaluated based on their weight change. First, cylindrical hydrogel samples (diameter × height = 5 × 2 mm) were prepared and air-dried to measure the dry weight (***W***_dry_) of the samples. The samples were then immersed in DW at room temperature, and the weights of wet samples (***W***_wet_) were measured at certain intervals. All measurements were performed in triplicate and the swelling ratio (%) was calculated using the following formula:(1)$$Swelling\;ratio\;\left(\%\right)=\frac{W_{wet}-W_{dry\;}}{W_{dry}}\times 100\%$$

Temperature and pH-responsive degradation studies of hydrogels were evaluated by the dry weight ratio of hydrogel samples. Well-established cylindrical KFH-3 and FIB-H hydrogels (diameter × height = 5 × 2 mm) were soaked in cell media containing plasmin and non-plasmin proteases, prepared at 5.5 pH and incubated at 50 °C, under mild shaking conditions. The hydrogels were taken at different time points (0.5, 1, 2, 3, 4, 5, 12, 24 h) and dried. The dried hydrogels were weighted (***W***_t_) and compared with the initial weights (***W***_0_) of samples. The weight change of hydrogels was calculated as follows:(2)$$Weight\;\left(\%\right)=\frac{\left(w_{o}-w_{t}\right)}{w_{o}}\times 100$$

### 3.8. Cell Culture

Primary HGF cells were purchased from ScienCell Research Laboratories. Cells were propagated in fibroblast medium, supplemented with 2% (*v*/*v*) fetal bovine serum, fibroblast growth supplement, and penicillin (100 U/mL)-streptomycin (100 μg/mL), on poly-l-lysine–coated flasks (ScienCell Research Laboratories, Carlsbad, CA, USA). The cells were cultured in a humidified incubator at 37 °C and 5% CO_2_. The culture medium was replaced every two days. HGF passages 4–7 were used in the experiments.

### 3.9. Cell Encapsulation

The KFH-3 was evaluated by the encapsulation test of HGF cells using the FIB hydrogel as a control. Briefly, HGF cells were suspended in media at a density of 5 × 10⁵ cells/mL. The cell suspension (200 μL) was then mixed with the same volume of FIB solution, or KRT-FIB solution with THR. After gel formation, 2% (*v*/*v*) fetal bovine serum, fibroblast growth supplement and penicillin (100 U/mL)-streptomycin (100 μg/mL) was added. Then, HGF-embedded hydrogels were maintained at 37 °C and 5% CO_2_ in a humidified incubator until processing.

### 3.10. Cell Viability

A LIVE/DEAD assay was used to investigate the viability of encapsulated HGF cells. After encapsulation, the samples were incubated in a solution of calcein-AM/ethidium homodimer for 30 min at 37 °C, immediately and after 3 days of culture. A calcein-AM/ethidium homodimer LIVE/DEAD assay was used to quantify cell viability according to the manufacturer’s instructions. Cell morphologies were observed under a fluorescence microscope (IX71; Olympus Life Science; Tokyo, Japan). The cell proliferation was evaluated using Cell Counting Kit-8 (CCK-8, DOJINDO, Kumamot, Japan), after incubation of 1, 3, 5, and 7 days. Briefly, the cultured cells were incubated with 10% CCK-8 working solution in a cell culture medium for 3 h at 37 °C, in the dark. Then, 100 μL of the supernatant was extracted to a new 96-well plate and the absorbance at 450 nm was measured using Plate Readers (PerkinElmer, MA, USA).

### 3.11. Confocal Laser Scanning Microscopy

HGF cells were seeded into the hydrogels and placed in 6-well plates. After 5 days of culture, the cells were encapsulated in a fixed hydrogel and stained with rhodamine-phalloidin (Invitrogen) and 2-(4-aminophenyl)-1H-indole-6-carboxamidine to visualize F-actin filaments and cell nuclei, respectively. Fluorescence images were obtained using an LSM 980 with Airyscan 2 (Carl Zeiss, Jena, Germany). The imaging setup comprised a water immersion lens with a 40× objective, a slice thickness of 5 μm, and a total thickness of 205 μm. The hydrogel images of the entire volume were obtained using fluorescence emission intensity, processed, and combined into a 3D volume using image processing software (ZEN black software, Oberkochen, Germany).

### 3.12. Statistical Analysis

Statistical analyses were carried out by one-way analysis of variance with Tukey’s post hoc analysis for significance. Statistical significance was set at *p* < 0.05. For rheology and image analysis, statistical significance was analyzed using the Origin software (Origin Software, CA, USA).

## 4. Conclusions

In this study, the injectable KFH was developed for improving the porosity and viscosity by controlling the molar ratio of KRT and FIB. The resulting KFH showed the synergistically improved proliferation of encapsulated HGFs, and it was also observed that HGFs exhibited a diffuse morphology when they were encapsulated in KFH. These results suggest that the novel KRT-based hydrogel has great potential for use as a scaffold for tissue regeneration for biomedical applications.

## Figures and Tables

**Figure 1 ijms-22-13269-f001:**
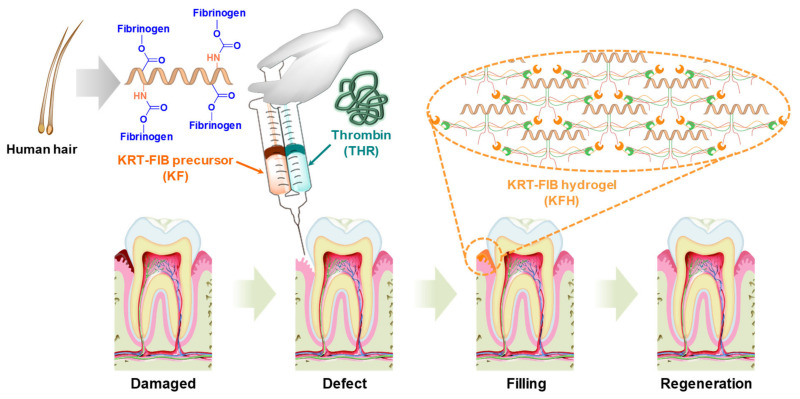
Schematic illustration of keratin-conjugated fibrinogen (KRT-FIB) hydrogels (KFHs) for oral tissue regeneration.

**Figure 2 ijms-22-13269-f002:**
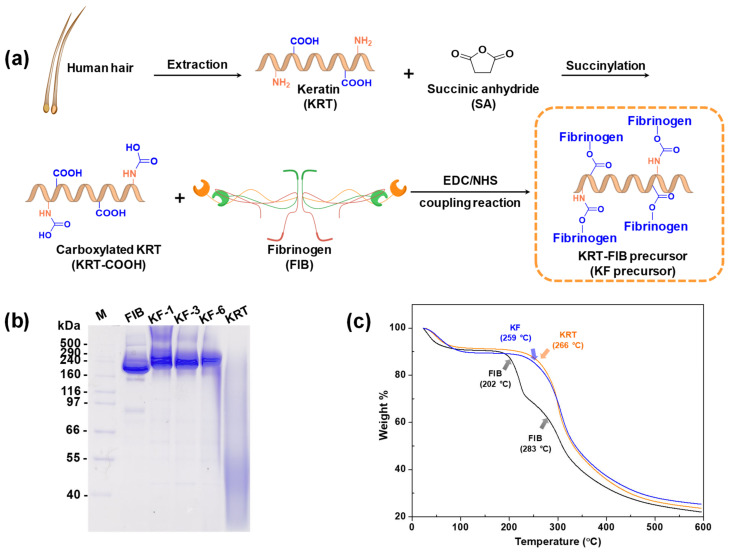
(**a**) Synthetic scheme of KRT-FIB precursor. (**b**) Thermogravimetric analysis diagrams of KRT, FIB, and KRT-FIB precursors. Each arrow indicates a temperature where the major weight loss starts. (**c**) Sodium dodecyl sulfate-polyacrylamide gel electrophoresis results obtained for the KRT; FIB; and KF-1, 3, 6 precursors.

**Figure 3 ijms-22-13269-f003:**
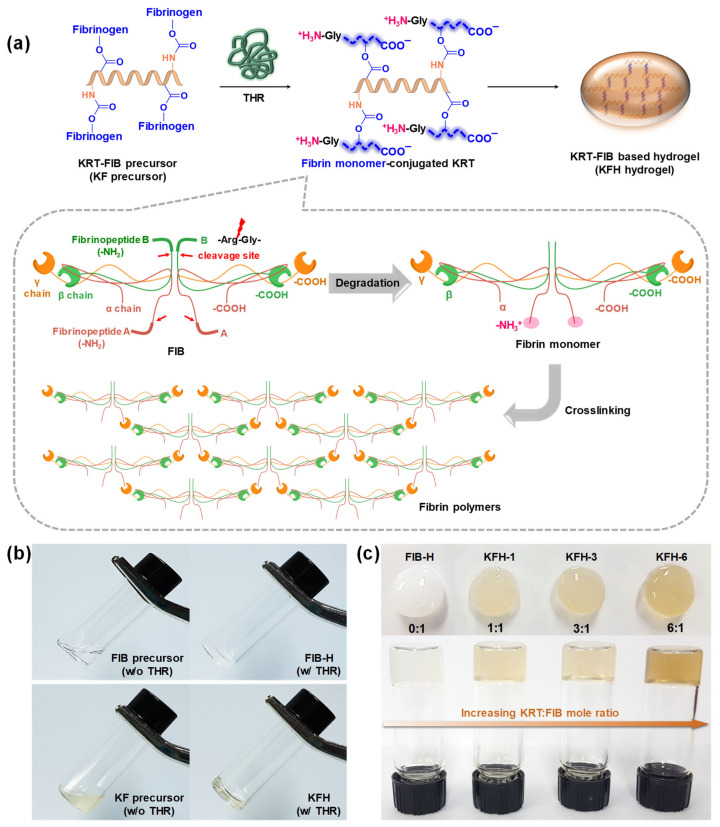
(**a**) Schematic illustration of KFH preparation and the mechanism of thrombin (THR)-induced fibrin cross-linking. (**b**) Gelation images of FIB and KRT-FIB precursors with or without THR. (**c**) Digital pictures of KFH samples with varying KRT:FIB mole ratios. White-colored turbid FIB-based hydrogel (FIB-H) gradually becomes brownish and transparent as the KRT content in the hydrogels increases.

**Figure 4 ijms-22-13269-f004:**
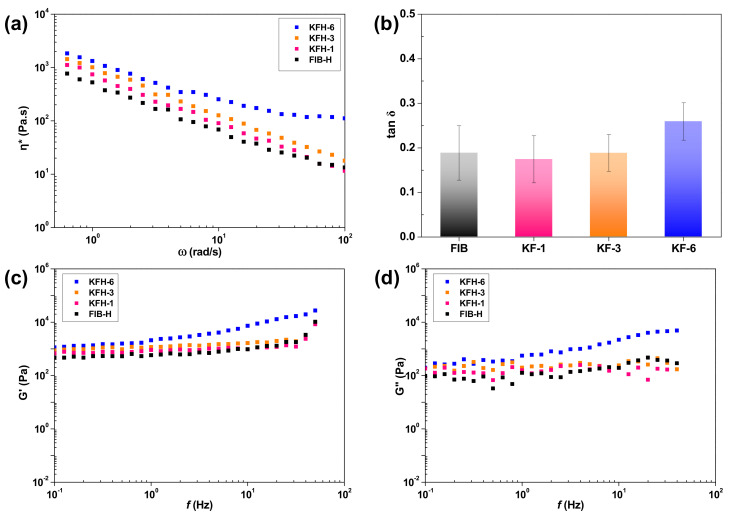
Characterization of THR-induced hydrogels: (**a**) Gelation images of hydrogels with different KRT compositions. Changes in (**b**) complex viscosity following the increase in frequency sweep, (**c**) storage modulus, and (**d**) loss modulus following the frequency. All measurements are performed with FIB-Hs as controls.

**Figure 5 ijms-22-13269-f005:**
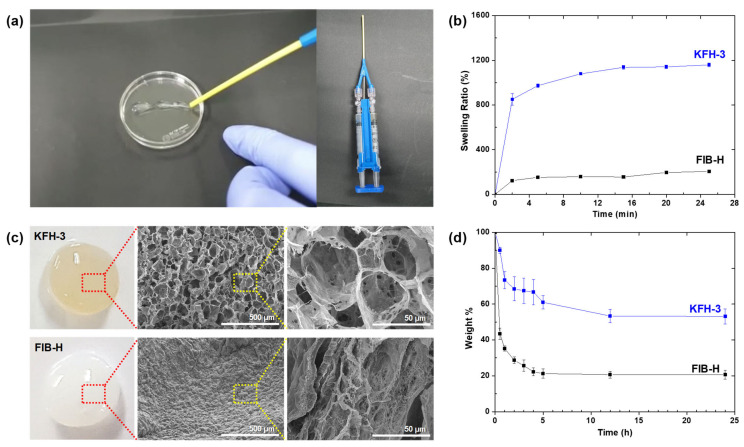
(**a**) The KFH-3 hydrogel was able to pass through a needle without clogging, indicating its high injectability (**left**); the precursor solution was loaded onto one side of a dual-chambered syringe with an 18 G needle (diameter = 1.3 mm), and the THR solution was loaded onto the other side (**right**). (**b**) Swelling kinetics of KFH-3 and FIB-H in deionized water at 37 °C. (**c**) Digital pictures and SEM images of KFH-3 and FIB-H at different magnifications (scale bar = 500 μm and 50 μm, respectively). (**d**) Temperature- and pH-responsive degradation study of hydrogels was evaluated by the dry weight ratio of the hydrogel samples.

**Figure 6 ijms-22-13269-f006:**
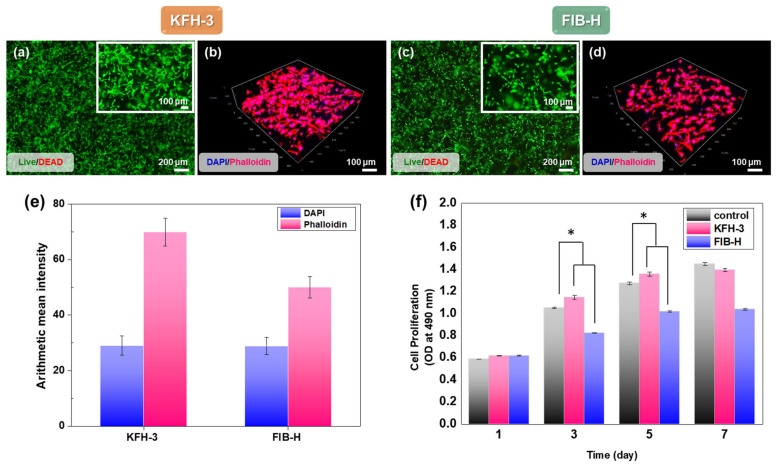
Characterization of cell adhesion and proliferation in three-dimensional cell encapsulation in FIB-H and KFH-3. For the cell viability assay, human gingiva fibroblasts (HGFs) embedded in FIB-H and KFH-3 were stained with calce-in_Am (green)/ethidium homodimer (red). LIVE/DEAD assay 24 h after encapsulation (**a,c**) shown at low (scale bar = 200 µm) and high (scale bar = 100 µm) magnification. (**b**,**d**) Confocal pictures of encapsulated HGF cells labeled with 2-(4-aminophenyl)-1H-indole-6-carboxamidine (DAPI) and Phalloidin (F-actin) (scale bar = 100 µm). (**e**) Expression of DAPI and phalloidin was quantified by immunofluorescence after 3 days in culture. (**f**) The proliferation of HGF cells encapsulated in KFH-3 and FIB-H for 7 days, as measured in a CCK-8 assay. Both data were evaluated for cyto-compatibility at a concentration of 20 mg/mL. Data are presented as the mean ± SD of triplicate experiments: * *p* < 0.05.

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
