# Peer review of "Injectable Human Hair Keratin–Fibrinogen Hydrogels for Engineering 3D Microenvironments to Accelerate Oral Tissue Regeneration"

_ijms, 2021, doi:10.3390/ijms222413269_

Round 1

Reviewer 1 Report

The manuscript can be improved by making the following changes:

-add a subsection under materials and methods, describing how the rheological and injectable testing was performed.

-In section 2.3, the initial few lines should come under methods section.

-Figure 5 in line 193 should be replaced as figure 5c.

-in line 222, mention what is considered to be the most ideal porosity for HGFs and how does the hydrogel compare to that. eg: pore size, pore density.

-Correct labeling in fig 6 and add error bars to fig 6g. 

-section 3.2 should be moved after the subsection on preparation of KFH. 

-mention how many replicates were used in the cell studies. 

-include some FTIR data to confirm the composition of hydrogels.

Reviewer 2 Report

The manuscript deals with an interesting topic and offers interesting research ideas. However, in my opinion in this form it seems to be preliminary. Please find below specific comments and major issues that must be addressed to reconsider the manuscript for publication.

Please improve the English throughout all the manuscript.

Introduction:

Authors should pay attention to the many repetitions. Also, the sentences in lines 47-48 (In general, biomaterials using natural polymers are highly biocompatible with cells and the surrounding tissues and are less inflammatory) is too generic. In fact, there are natural polymers that induce an inflammatory response when implanted in the body (e.g. fibroin). Less inflammatory compare to what?

Biocompatible with the cells= cytocompatible

Please rewrite.

Results and discussion:

  • Lines 105-107:Although both KRT and FIB are hydrophilic proteins, they cannot be integrated into homogeneous hydrogels in the presence of THR. To address this challenge, covalently 106 linked KRT-FIB precursors were synthesized via a facile coupling reaction.

A more in-depth discussion is needed. Please explain in detail the reason is impassible to mix homogenously the proteins in the hydrogel

  • The time of gelation should be studied performing rheological studies evaluating G’ and G’’ values as a function of time. for all the other rheological studies no experimental details are reported. Was the linear viscoelastic region determined for each hydrogel prior to perform the experiment? What was the strain % used? How was chosed?
  • Since hydrogels are produced by the crosslinking mediated by the presence of FIB, is it correct to compare the properties of hydrogels containing different amount of this protein? Hydrogels produced with the same amount of precursors contain different amount of FIB. Please explain.
  • The work rationale is that by conjugating KRT to FIB it is possible obtain a biomaterial able to produce hydrogels that overcome the problem related to the stiffness of fibrin hydrogel.

There no evidences or results in the manuscript that support this hypothesis.

On the contrary, the viscoelastic properties of the hydrogels obtained by the hybrid material show a higher elastic modulus G’ compare to the FIB hydrogel. Please explain

  • What is the degradation profile of the obtained hydrogel? This aspect must be taken in consideration as well as the possibility to measure the mechanical properties of the obtained hydrogels.
  • Authors stated that KFH has a porosity higher than 95%. Please explain how it was calculated.
  • The biological experimental design is poor.
  • The proliferation must be measured compared the cells viability as a function of culture time by means of of MTT, MTS, Alamar blue and so on…
  • From the results showed it is not possible to appreciate any significative differences between KFH and FIBH.

Round 2

Reviewer 2 Report

there are stil some points that authors have not addressed.

Please see specific comment below:

  • Please insert the detailed description of the experimental procedure (that reported in the responses) for the rheological analyzes in the main text.
  • The method reported for the calculation of porosity (using ImageJ software) is not reliable. Porosity is a bulk material feature and must be calculated by means of BET isotherm method. I suggest to eliminate this result from the manuscript (at least authors should furnish references that use this method).
  • It is not sufficient to explain the degradation mechanism of the starting polymers. The degradation profile should be reported as example, as of % of recovered weight as a function of incubation time. This experiment is important to corroborate your rationale.
  • Cell proliferation is demonstrated showing that cells viability increases over incubation time. This cannot be demonstrated from the reported results.

Round 3

Reviewer 2 Report

Authors addressed the major issues improving the quality and the clarity of the manuscript. I suggest to insert in the manuscript results of cells proliferation. In my opinion, in this form the manuscript is suitable for publication.